# OpenReview forum: "Monitoring Decomposition Attacks with Lightweight Sequential Monitors"
_ICLR.cc/2026/Conference — ICLR 2026 Poster_

### Official Review · Reviewer_eW3J · 2025-10-29

**Soundness:** 3
**Presentation:** 2
**Contribution:** 3
**Rating:** 6
**Confidence:** 4

**Summary:**

This paper shows that well aligned models such as GPT-4o and Llama-Guard are vulnerable to decomposition attacks, and are effective across different settings. It proposes that lightweight models can cumulatively monitor the queries of decomposition attacks as they progress, and through careful prompt engineering, outperform much stronger baselines in terms of detection rates and cost. The paper also introduces an extensive new dataset for decomposition attacks, where each task is broken down seemingly benign subtasks.

**Strengths:**

1. The paper is timely, as this problem is a growing concern.
2. The solution is simple, cheap and effective, beating far more expensive zero-shot baselines.
4. The evaluation is extensive, the metrics used are appropriate and the results are convincing.
3. The dataset will prove very useful to the area going forward as a benchmark.

**Weaknesses:**

1. The defense is reliant on an engineered, static prompt to control detection behavior, which raises concerns about adaptive attacks. While the PAIR baseline does give an adaptive attack (where subtasks are made more benign while maintaining semantics), it doesn't include the system prompt as part of the input, which can lead to a suboptimal objective for the adversary.

2. The dataset could more details and descriptions regarding the diversity of tasks (subcategories), length of decomposed prompts, whether the subtasks are independent of each other, etc.

**Questions:**

1. How well does the sequential monitor framework perform when the ICL prompt is included as part of PAIR's input?

2. Are there scenarios where prompts can be decomposed into independent subtasks? Adversaries could make singular independent queries to avoid providing a cumulative prompt history to the monitor.

---

> ### Author Response · Authors · 2025-11-16
>
> Thank you for engaging with our work and for the positive rating. We answer your questions below:
>
> > Weakness 1: The defense is reliant on an engineered, static prompt to control detection behavior, which raises concerns about adaptive attacks. While the PAIR baseline does give an adaptive attack (where subtasks are made more benign while maintaining semantics), it doesn't include the system prompt as part of the input, which can lead to a suboptimal objective for the adversary.
>
> > Q1: How well does the sequential monitor framework perform when the ICL prompt is included as part of PAIR's input?
>
> Thank you for raising this point. Per your request, we ran an experiment in which the automated red team (PAIR) knows the exact monitor prompt and iteratively rewrites subprompts to bypass it (we provide full details in Section 5.4, highlighted in yellow). The TL;DR is that our defense method remains robust, and as the iteration budget increases, our monitoring performance plateaus at approximately F1 = 0.90 and DSR = 0.87.
>
> > Weakness 2: The dataset could more details and descriptions regarding the diversity of tasks (subcategories), length of decomposed prompts, whether the subtasks are independent of each other, etc.
>
> We appreciate this excellent suggestion. We now include the decomposed prompt lengths in Table 6, the category distribution in Table 7, and our manual checking results for subtask independence in Appendix A.
>
> > Q2: Are there scenarios where prompts can be decomposed into independent subtasks? Adversaries could make singular independent queries to avoid providing a cumulative prompt history to the monitor.
>
> Thank you for the question. We note that in Lines 11 to 13, our motivation here is the agentic scenario: as LLMs become more agentic, users assign “stateful” instructions (whether it’s QA, text-to-image, or general-agent instruction) to agents, where everything happens in the same context. We further emphasize this point in Lines 57 to 59, that “as LLMs become more agentic, they pose a realistic, high-impact risk, where the harm could be decomposed sequentially”. The "sequential" means the instructions are stateful. In such stateful settings, the monitor has access to the full prompt history.
>
> We agree that a fully stateless setting, where an adversary issues isolated, independent queries, is also important. As noted in Lines 495–496, we acknowledge this independent, stateless case as a limitation. We then discuss this further in the newly added "future work" paragraph, where we outline ways to address stateless decomposition attacks through policy formation, such as requiring government-issued ID tied to each user account, such that if they are caught conducting malicious activities, they jeopardize their future access to the model. On top of that, on the technical side, frontier AI labs can potentially address the stateless setup by chaining semantically relevant inputs from different contexts into a single thread per user account (for example, by embedding each input and calculating cosine similarity to select the most similar ones), which can then be monitored by our lightweight sequential monitor in one context.
>
> Moreover, as stated in our conclusion, our evaluation intentionally focuses on stateful, single-trajectory monitoring for agentic LLMs, which we view as a critical, and currently underexplored, risk surface as these LLMs become increasingly more agentic.
>
> Additional note: Given that Anthropic’s recent report on an AI-orchestrated cyber-espionage campaign [https://www.anthropic.com/news/disrupting-AI-espionage] explicitly uses decomposition attacks, “broke down their attacks into small, seemingly innocent tasks that Claude would execute without being provided the full context of their malicious purpose”, we think our work is highly relevant, realistic, and well-positioned to address this increasingly effective misuse strategy.
>
> Please let us know if you have any other questions that we can address. If not, we sincerely hope you can consider increasing your rating to 8 (accept), as we believe that our contribution is significant to the safety community that aims to guard against sophisticated, realistic misuse.

---

> > ### Comment · Reviewer_eW3J · 2025-11-20
> >
> > Thank you for running the additional experiment and including these new details! I have raised my score to 8.

---

> > > ### Author Response · Authors · 2025-11-20
> > >
> > > We appreciate you updating the score to an 8. Thank you for acknowledging our contribution.

---

### Official Review · Reviewer_JNS6 · 2025-11-01

**Soundness:** 2
**Presentation:** 3
**Contribution:** 2
**Rating:** 4
**Confidence:** 4

**Summary:**

The paper examines decomposition attacks against LLM agents. These involve breaking down malicious intents into subtasks that are safe when taken in isolation and can bypass safety filters.

The paper presents DecomposedHarm, a dataset of 4634 human-annotated agent tasks for conducting and evaluating such attacks, and finds that they are up to 87% successful against GPT-4o.

The authors also develop a defense using sequential monitoring of LLMs which can detect these attacks with up to 93% success and with low additional latency.

**Strengths:**

1. The research problem holds significant value, as attacks decomposed through simple prompts remain undetectable by LLMs. This issue exhibits universality, scalability, and urgency, while the novel dataset DecomposedHarm demonstrates high research value.
2. The author innovatively introduces a defense method achieving a maximum defense success rate of 93%, while maintaining robustness under adversarial conditions.
3. DecomposedHarm is an extensive and diverse dataset for studying decomposition attacks, providing clear splits (Table 4) and strong empirical visualization (Figure 2).
4. The authors provides solid quantitative analyses (Figures 2&5, Tables 1–3), consistently showing decomposition sharply reduces refusal rates and generalizes across models and interaction modes.

**Weaknesses:**

1. Applying o3-mini, GPT-4o, and Claude-3.7-Sonnet model can reach the highest F1 scores, but the costing problem is also concerning, especially examing cumulative context.
2. The authors only applies in-context learning (prompt engineering) methods to improve the sequential monitors.
3. The ability of defense in the method does not stem from the robustness of the algorithm itself, but rather from the accidental effect of the prompt wording.
4. Although the decomposition attack (Fig. 9) demonstrated in the paper is effective, it is limited to non-adaptive scenarios and single pipelines.

**Questions:**

1. How much does monitoring performance drop if the best-performing ICL or CoT prompts are replaced with simpler, less-engineered prompts? Can the authors quantify the risk that monitoring is reliant on specific prompts?
2. If a single attack prompt can be decomposed into many sub-questions, achieving high F1 could bring significant token consumption. How do the authors address this scalability issue?
3. Please report the distribution of harmful metric percentages (harmful metric / total subtask length) and stratify performance reporting by this percentile in test evaluations. Fail to do so may introduce bias in assessments due to the tendency to place harmful steps in advantageous positions.
4. Please evaluate the framework under varying benign-subtask injection rates by fixing the fraction of benign subtasks (e.g., 25%, 50%, and 75% ,benign subtasks / total subtasks). For each setting report F1, cost per task, and average latency. No need for generating new tasks, just pick the tasks that satisfies corresponding ratio already had.

If the main concerns have been addressed, I am considering raising my score.

**Details Of Ethics Concerns:**

The authors claimed that the work has examples that may be considered harmful or offensive, therefore ensuring these examples can be used for research is essential.

---

> ### Author Response · Authors · 2025-11-16
>
> Thank you for reviewing our work and for the detailed comments. We address your questions below:
>
> > Weakness 1: Applying o3-mini, GPT-4o, and Claude-3.7-Sonnet model can reach the highest F1 scores, but the costing problem is also concerning, especially examing cumulative context.
>
> Thanks for this comment. We note that these costly models serve only as baselines and are not part of our proposed method (with cheap models like GPT-4o-mini, GPT-4.1-nano, and Llama-3.1-8B). As stated in lines 255–258, these strong but costly baselines represent unrealistic choices and therefore function solely as upper-bound references that our method aims to surpass. For this reason, we are unsure why this is a weakness, and we appreciate any clarification.
>
> > Weakness 2: The authors only applies in-context learning (prompt engineering) methods to improve the sequential monitors.
>
> We appreciate this feedback. We note that this is because using the ICL method already works surprisingly well, and there’s no need for advanced techniques. In fact, this is the point of our paper: to show that an easy approach can surpass strong reference models such as o3-mini, GPT-4, and even GPT-5 in monitoring the effective decomposition attack. Moreover, simplicity is an advantage because (1) it doesn’t require extensive training that consumes any compute, (2) it’s very easy to implement.
>
> > Weakness 3: The ability of defense in the method does not stem from the robustness of the algorithm itself, but rather from the accidental effect of the prompt wording.
>
> Thank you for this comment. We note that this is not an accidental effect; we evaluate three different lightweight models and show in Table 1 that all of them achieve significant performance gains with optimized prompts. Additionally, we mostly employ the prompting techniques that prior work have *empirically* shown are effective. For example: ICL examples [Brown et al.] and CoT prompting [Wei et al.]. Lastly, we believe that these models can even be further improved using a GEPA-like approach [Agrawal et al.].
>
> -Language Models are Few-Shot Learners. Brown et al. (2020)
>
> -Chain-of-Thought Prompting Elicits Reasoning in Large Language Models. Wei et al. (2022)
>
> -GEPA: Reflective Prompt Evolution Can Outperform Reinforcement Learning. Agrawal et al. (2025)
>
> > Weakness 4: Although the decomposition attack (Fig. 9) demonstrated in the paper is effective, it is limited to non-adaptive scenarios and single pipelines.
>
> We appreciate this feedback. We explicitly explore the adaptive scenario in Sec 5.3, where the adaptive attacker (automated red team) iteratively rewrites subtasks to be more benign whenever harmful intent is detected, with up to 50 iterations in the budget, yet our monitor remains robust. Specifically, we see that as the iteration budget increases, the defense success rate and F1 score plateau at high values (F1: 0.87, DSR: 0.81). Moreover, in Sec 5.4, we add another scenario in which we give the adaptive attacker an additional advantage: they know the exact monitor prompt the monitor is using, such that they can iteratively revise their subprompts to bypass the monitor. Nevertheless, we find that our monitor still remains robust, and the defense success rate and F1 score plateau at high values (F1: 0.9, DSR: 0.87). These results suggest that, under a strong adaptive scenario, our proposed defense method remains effective.
>
>
> > Q1: How much does monitoring performance drop if the best-performing ICL or CoT prompts are replaced with simpler, less-engineered prompts? Can the authors quantify the risk that monitoring is reliant on specific prompts?
>
> We appreciate this question. We conduct extensive prompt evaluations in Sec. 4.1. Specifically, Table 7 shows that across the 5 prompting techniques we tested, all of them outperform the baseline binary prompt (which simply asks for a True/False output) for most of the 5 (advanced prompts) × 3 (models) combinations. (The only exception is the safety-guideline setup for GPT-4.1-nano) In practice, model developers would perform far more extensive prompt evaluations before deployment and would likely identify an even stronger prompt. Therefore, we respectfully note that this should not be a concern.
>
> > Q2: If a single attack prompt can be decomposed into many sub-questions, achieving high F1 could bring significant token consumption. How do the authors address this scalability issue?
>
> Thank you for raising this point. We explicitly examine scalability in Sec. 5.2, where we stress-test our monitor under scenarios with a large number of subtasks (up to 160 subtasks per original task, compared to roughly 10 subtasks in the original tasks). We show that the monitor remains robust even when the subprompt list becomes extremely long, and that as the number increases, monitoring performance plateaus at a high value (0.8 F1/DSR) rather than degrading.

---

> > ### Author Response · Authors · 2025-11-16
> >
> > > Q3: Please report the distribution of harmful metric percentages (harmful metric / total subtask length) and stratify performance reporting by this percentile in test evaluations. Fail to do so may introduce bias in assessments due to the tendency to place harmful steps in advantageous positions.
> >
> > Thank you for this good suggestion. Because we did not save the low-level metric, we thus invested significant efforts in rerunning our evaluation to answer your question. We report all the findings in Appendix Section F.1 on page 25. The tl;dr is that across all harmful index position percentile bins, our optimized lightweight sequential monitor, including GPT-4o-mini and Llama-3.1-8B, outperforms all reference models, including GPT-4, o3-mini, and GPT-5.
> >
> > > Q4: Please evaluate the framework under varying benign-subtask injection rates by fixing the fraction of benign subtasks (e.g., 25%, 50%, and 75% ,benign subtasks / total subtasks). For each setting report F1, cost per task, and average latency. No need for generating new tasks, just pick the tasks that satisfies corresponding ratio already had.
> >
> > Thank you for this good idea as well. Per your request, we conducted this analysis to examine the model's performance while fixing the fraction of benign subtasks. We report all the stats in Appendix Section F.2 on page 25. TL;DR: across all injection rates and benign fractions, our lightweight sequential monitor, GPT4o-mini, still outperforms strong baselines, including GPT-4, o3-mini, and GPT-5, in F1, cost per task, and average latency.
> >
> > Additional note: Given that Anthropic’s recent report on an AI-orchestrated cyber-espionage campaign [https://www.anthropic.com/news/disrupting-AI-espionage] explicitly uses decomposition attacks, “broke down their attacks into small, seemingly innocent tasks that Claude would execute without being provided the full context of their malicious purpose”, we think our work is highly relevant, realistic, and well-positioned to address this increasingly effective misuse strategy.
> >
> > Please let us know if you have any other questions that we can address. If not, we sincerely hope you can consider increasing your rating to 8 (Accept) or 6 (Borderline Accept), as we believe that our contribution is significant to the safety community that aims to guard against realistic, sophisticated misuse.

---

> > > ### Comment · Reviewer_JNS6 · 2025-11-17
> > >
> > > Thanks a lot for your clarification. I've already seen the supplementary analysis on the distribution of harmful metric percentages in the new paper version. This distribution analysis is indeed crucial, and its inclusion has elevated the experimental results to the expected level of credibility. It's now clear that employing more economy-friendly models achieves higher and more stable F1 scores and latency across arbitrary harmful metrics. Although most tasks incorporate the harmful index at the very last step, I consider this a minor issue that's nice to have addressed. The clarifications on prompt sensitivity, scalability, and benign-subtask ratios enhance my confidence in the paper's quality. Given these improvements, I am updating my score to 6.

---

> > > > ### Author Response · Authors · 2025-11-17
> > > >
> > > > Thank you for acknowledging our contribution and for raising the score to 6. We appreciate this.

---

> ### Author Response · Authors · 2025-11-19
>
> Hello, we noticed that you flagged for ethics review and mentioned: “The authors claimed that the work has examples that may be considered harmful or offensive, therefore ensuring these examples can be used for research is essential.”
> To clarify, our intention is to highlight how harmful current LLMs can be, which is why we included these harmful/offensive examples. Our goal is to raise awareness and seriousness of the issue and present a defense method to mitigate such behavior.
> The last thing we want is for any examples to unnecessarily disturb readers. Could you let us know if there are specific examples you find particularly unacceptable so that we can revise or remove them? For instance, we wonder whether you feel Figure 4 is overly explicit, and we can blur or remove it if needed. Please let us know, and we will revise it right away.

---

> > ### Comment · Reviewer_JNS6 · 2025-11-19
> >
> > Overall, I find the paper solid and the contributions meaningful. Regarding Figure 4, I recommend removing it. Removing this figure would not affect the scientific quality of the paper and would make the presentation more appropriate.

---

> ### Author Response · Authors · 2025-11-19
>
> We appreciate the feedback, and we have now removed Figure 4.
>
> Please let us know if you have any other questions that we can address. If not, we sincerely hope you can consider removing the ethics review flag.

---

> > ### Comment · Reviewer_JNS6 · 2025-11-19
> >
> > Sure, I’ve removed the ethics review flag, as the sensitive content has now been addressed.

---

### Official Review · Reviewer_svdc · 2025-11-03

**Soundness:** 2
**Presentation:** 2
**Contribution:** 2
**Rating:** 2
**Confidence:** 3

**Summary:**

The paper looks at decomposition attacks in the context of LLM agents. These are attacks where a harmful task is broken up into a sequence of benign steps that are cumulatively harmful. The authors produce datasets that capture this category of jailbreaks and then introduce sequential monitors that can detect decomposition attacks. They demonstrate that prompt optimization of a sequential monitor can block decomposition attacks more effectively than standard filters.

## Contributions

 * A novel dataset of decomposition attacks
 * A sequential monitoring method for filtering
 * Results that indicate their sequential monitoring is feasible with reasonably small monitoring models

**Strengths:**

Overall, the paper deals with an important area of concern: decomposition attacks are an effective attack and mitigations are an important area for further study. As a result, they are studying a problem of interest to the community.

The dataset contribution is the most valuable from my perspective, although prior work has defined and demonstrated decomposition attacks, I believe there is no existing dataset of decomposition attacks that allows for comprehensive evaluation or study.

Separately, the approach described for defense seems reasonable, and the authors do a good job of balancing concerns for the effectiveness of monitors with the feasibility of deployment.

In summary, the paper studies an important problem and provides a reasonable defense proposal. The evaluations indicate that this is a potentially promising direction for the development of new guardrails.

**Weaknesses:**

The primary weakness of the paper is a heavy level of overclaiming. The introduction and title suggest that it is "surprisingly easy" to defend against these attacks. However, I do not believe the author's evaluations are sufficient to justify such a claim. There are a few specific issues with the framing:

 * The authors study a variant of the problem where the decomposition attack occurs within a single context. This is easily bypassed if the attacker can carry state over from one context to another. For example, they describe an image generation task to generate a potentially racist propaganda image that violates guidelines. However, it seems like it would be straightforward to split these requests into several contexts and avoid the defense mechanisms proposed. While this is briefly mentioned in the limitations section, it deserves further discussion and motivation for why this particular attack setting is appropriate for study.

 * While the authors do some study of adaptive attacks, it is not clear that these are representative of how a capable attacker would behave. In particular, 10 attempts by GPT-4.1 is a valuable measurement, but far below the level of resources that a capable attacker could apply to such a system.

 * Most of the comparisons show performance against o3-mini as an expensive evaluator; they don't show that performance is good enough from an objective risk minimization perspective. As a result, the evidence the authors actually have supports that lightweight sequential classifiers can be optimized through prompt optimization to perform similarly to more expensive classifiers. This is a different and weaker claim.

 * It seems possible that there is information leakage from the similarity in how the data is generated. There's an attempt to account for this by using in-context examples from one task and exploring how well it transfers to another. I'm not sure what justifies this approach. Similarly, the fact that the examples from the train/validation set are programmatically selected by the same process as the test set means that this approach risks undercounting vulnerability.

## Clarity about evaluations

I also have some questions about the evaluation setup overall.

 * First, in comparison to GPT-4o and o3-mini as a reference,  why are the prompts unoptimized? It seems to me that it would be appropriate to compare to an optimized prompt for the reference models as well. This provides an effective limit on performance. By comparing against an unoptimized prompt for the more expensive models, you are stacking the deck in your favor. This makes the analysis of the results misleading: the optimized sequential monitor matches the performance of a more expensive, but unoptimized model.

 * Second, it's unclear why the authors are comparing 4.1-nano with 4o-mini. This is a bit strange because they are different model sizes and I would expect 4.1 to be more effective as a guardrail model. This raises some questions about the overall choices of which models to compare and why they were chosen over alternatives.

**Questions:**

* How did you evaluate the quality of your LLM-judge for harmfulness?
 * How did you select which models to use for your experiments? E.g., why is 4o-mini being compared with 4.1-nano?
 * Is there a way to evaluate your performance against a better-resourced adversary?
 * Can you explain why these defenses are interesting, given that we can/should expect decomposition attacks to be executed across different contexts?
 * Can you clearly and concisely articulate how the evidence in your paper supports the claim that defending against decomposition attacks is 'surprisingly easy'?

---

> ### Author Response · Authors · 2025-11-16
>
> Thank you for reviewing our paper and for the thoughtful comments. We answer your questions below:
>
> > Weakness 1: The authors study a variant of the problem where the decomposition attack occurs within a single context. {...}
>
> > Q4: Can you explain why these defenses are interesting, given that we can/should expect decomposition attacks to be executed across different contexts?
>
> Thank you for the comment on weakness 1 and Q4. We note that in Lines 11 to 13, our motivation here is the agentic scenario: as LLMs become more agentic, users assign “stateful” instructions (whether it’s QA, text-to-image, or general-agent instruction) to stateful agents, where everything happens in the same context. We emphasize this point again in Lines 57 to 59, that “as LLMs become more agentic, they pose a realistic, high-impact risk, where the harm could be decomposed sequentially”. The "sequentially" means the instructions are stateful. In these settings, an attacker can sequentially decompose a harmful goal while preserving previous instructions, exactly the scenario we study.
>
> Additionally, as you mentioned, we acknowledge the stateless case as a limitation in our Limitation section. We then discuss this further in Appendix E.1, where we outline ways to address stateless decomposition attacks through policy formation, such as requiring government-issued ID tied to each user account, such that if they are caught conducting malicious activities, they jeopardize their future access to the model. On top of that, on the technical side, frontier AI labs can potentially address the stateless setup by chaining semantically relevant inputs from different contexts into a single thread per user account (for example, by embedding each input and calculating cosine similarity to select the most similar ones), which can then be monitored by our lightweight sequential monitor in one context.
>
> Moreover, as stated in our conclusion and Lines 498-499, our evaluation intentionally focuses on stateful single-trajectory monitoring for agentic LLMs, which we view as a critical, and currently underexplored, risk surface as these LLMs become increasingly more agentic.
> Lastly, Anthropic recently reported an AI-orchestrated cyber espionage campaign [https://www.anthropic.com/news/disrupting-AI-espionage] in which the attackers specifically conducted decomposition attack: “They broke down their attacks into small, seemingly innocent tasks that Claude would execute without being provided the full context of their malicious purpose." This suggests that the problem we study, decomposition attacks, is highly relevant and realistic, and we believe our work can effectively help mitigate this in a cheap and efficient way.
>
> > Weakness 3: Most of the comparisons show performance against o3-mini as an expensive evaluator; they don't show that performance is good enough from an objective risk minimization perspective. As a result, the evidence the authors actually have supports that lightweight sequential classifiers can be optimized through prompt optimization to perform similarly to more expensive classifiers. This is a different and weaker claim.
>
> We appreciate this comment. We note that our evaluation is not limited to comparisons against stronger monitors. We also report the raw defense success rate (best DSR ≈ 93%) and F1 score (best is 0.918) of our defense method computed against *ground-truth risk labels*, as shown in Tables 1-3 and Figures 5-7. Notably, the “ground-truth risk labels”, or as we described as the “harmful index” are explained in Line 144 for our agent tasks, in lines 177 and 178 for the test-to-image tasks, and in line 210 for the QA task, and the evaluation framework using these harmful indices (ie how we calculate the TP, TN, FP, and NP), is precisely explained in the Sec 3.1. These metrics (DSR + F1) based on the ground-truth risk labels, therefore, reflect objective risk-minimization performance rather than merely comparisons with stronger monitors. (Additionally, we include GPT-5 (way stronger than o3-mini), and our monitor still outperformed it, as shown in Table 1.)
>
> > Weakness 4: It seems possible that there is information leakage from the similarity in how the data is generated. {...}
>
> Thank you for this great point. As you requested, we ran additional experiments to evaluate whether such information leakage issues exist. We report the detailed procedures and findings in Appendix A.2 on page 16. The TL;DR is that we observe only trivial differences in monitoring performance before and after duplicate removal, with all changes falling well within one standard error of the original estimates. This outcome is expected: our monitor relies on only 8 ICL examples, and the test set has a low proportion of duplicates.

---

> > ### Comment · Reviewer_svdc · 2025-11-18
> > **Thanks for the response, still hung up on stateful vs stateless**
> >
> > Thanks for the response. I feel like this addresses Weaknesses 3/4 reasonably well. If you can commit to making this more clear in the paper, I'd be satisfied here.
> >
> > I should be clear that I think stateful decomposition attacks are interesting. However, I think the paper is not nearly clear enough about this scope up front. For example, afaict, the attack you reference from Anthropic was an example of a stateless attack where the decomposition between attacks occurred across contexts. Is there something that I've missed?
> >
> > I find this particularly important given the framing that defending against this attack is "surprisingly easy". My read of the evidence is that defending against *stateful* decomposition attacks is feasible with acceptable cost. I think this could be resolved by better scoping of the claims around how hard defense is and/or a discussion of the scope of attacks considered earlier in the paper (e.g., somewhere in the introduction). General stateless decomposition attacks are *hard* at the current state of the art.

---

> ### Author Response · Authors · 2025-11-16
>
> > Q: First, in comparison to GPT-4o and o3-mini as a reference, why are the prompts unoptimized? It seems to me that it would be appropriate to compare to an optimized prompt for the reference models as well.  {...}
>
> We appreciate this question. Our intention is to compare baselines and lightweight monitors *under comparable and realistic cost constraints*. For the costly reference models such as GPT-4o, o3-mini, GPT-5, prompt optimization generally requires substantially longer, more complex prompts, which would make their inference costs and latency even higher than the already expensive default configuration. Because these models are not practical candidates for monitoring, further increasing their cost through optimization would not be a realistic deployment scenario. Our question is thus whether a lightweight, optimized monitor can match or exceed the practical performance level of these costly models when run in a cost-feasible configuration. As shown in Table 1 (page 7), even after prompt optimization, the lightweight sequential monitor remains far more cost-efficient than the reference models.
>
> > Q: Second, it's unclear why the authors are comparing 4.1-nano with 4o-mini. This is a bit strange because they are different model sizes and I would expect 4.1 to be more effective as a guardrail model. This raises some questions about the overall choices of which models to compare and why they were chosen over alternatives.
>
> > Q2: How did you select which models to use for your experiments? E.g., why is 4o-mini being compared with 4.1-nano?
>
> Thank you for these two questions. To clarify, we are not necessarily comparing GPT-4.1-nano to GPT-4o-mini; rather, we optimize each lightweight model independently. As stated in lines 266–268, we selected these models because they are the lowest-cost options among the nine we studied and therefore the most relevant for evaluating monitoring to achieve low cost in deployment. For completeness, the reference models were chosen because they are the three most expensive models, as you can see from Fig. 5 (a).
>
> > Q1: How did you evaluate the quality of your LLM-judge for harmfulness?
>
> We use the StrongReject judge (Souly et al., 2024) to evaluate harmfulness. As footnoted on page 4, Souly et al. show extensive human studies that StrongReject’s judgments closely align with human annotations (kindly refer to their Section 3 for details).
>
> > Weakness 2: While the authors do some study of adaptive attacks, it is not clear that these are representative of how a capable attacker would behave. In particular, 10 attempts by GPT-4.1 is a valuable measurement, but far below the level of resources that a capable attacker could apply to such a system.
>
> > Q3: Is there a way to evaluate your performance against a better-resourced adversary?
>
> Thank you for these two questions. We ran two additional experiments for you: (1) increasing the automated red team’s attack budget from 10 attempts to 50 (Section 5.3, highlighted in yellow), and (2) evaluating a scenario in which the automated red team knows the exact monitor prompt and iteratively rewrites subprompts to bypass it (Section 5.4, highlighted in yellow). Under both settings, our lightweight monitors remain robust. In experiment (1), performance plateaus at roughly F1 = 0.87 and DSR = 0.81 as the iteration budget increases. In experiment (2), performance similarly plateaus at approximately F1 = 0.90 and DSR = 0.87.
>
> Moreover, we’d like to note that our adversarial setup (Section 5) already models a high-effort attacker: after the initial harmful query fails, the adversary performs decomposition, and if that also fails, they further try one or all advanced strategies (benign-task obfuscation, task inflation, automated red teaming). These scenarios reflect a substantially resource-intensive and resilient adversary. Given the four diverse stress test scenarios we have already tested, we believe we provide a sufficiently rigorous assessment of monitor robustness.

---

> > ### Author Response · Authors · 2025-11-16
> >
> > > Q5: Can you clearly and concisely articulate how the evidence in your paper supports the claim that defending against decomposition attacks is 'surprisingly easy'?
> >
> > Yes. We can concisely articulate this claim through 3 findings: (1) our optimized lightweight monitor (e.g., GPT-4o-mini) achieves a 93% DSR and outperforms much stronger models such as GPT-5, GPT-4o, and o3-mini in performance, cost, and latency; (2) it remains robust across four challenging adversarial pressures—task injection, extreme subtask obfuscation, automated red teaming with a 50-iteration budget, and an adaptive red team with knowledge of the monitor prompt, yet our defense consistently maintaining F1 > 0.87 and DSR > 0.8; and (3) All of this is achieved using only prompt optimization with the sequential monitoring framework, without model retraining or architectural changes. Therefore, we claim that defending against decomposition attacks is 'surprisingly easy'.
> >
> > Please let us know if you have any other questions that we can address. If not, we sincerely hope you can consider increasing your rating to 6 (Borderline Accept), as we believe that our contribution is significant to the safety community that aims to guard against sophisticated misuse.

---

> > > ### Comment · Reviewer_svdc · 2025-11-18
> > > **Thanks, this is good**
> > >
> > > Thanks for the clarity here. I think this is convincing. I'd recommend making your claims more clearly grounded in this evidence to avoid misinterpretation. I would also be more comfortable if the claim is more carefully scoped to apply to *stateful* decomposition attacks *within a single context*.
> > >
> > > The discussion could more clearly lay out future work on extending to stateless attacks across distinct contexts.

---

> > ### Comment · Reviewer_svdc · 2025-11-18
> > **Thanks for engaging, some followups**
> >
> > I understand the desire to compare across appropriate settings, however I'd encourage you to compare with prompt-optimization for more capable models to understand the limits of performance. This would help understand if, e.g., distilling a prompt-optimized larger model through finetuning could be valuable. I'm sensitive to concerns about cost, but I want to suggest the direction as a valuable one.
> >
> > I'm still confused by 4.1-mini wasn't considered --- it seems like that would be an acceptable cost for an important setting and it would be valuable to know if it gives a performance boost over 4.1-mini. As I said above, I don't think this is necessary but it's worth considering.
> >
> > RE: judge validation, please make sure that this is clearly communicated in the paper. As a followup question, can you address the possibility of distribution shift in your queries reducing the alignment between the LLM judge and human evaluations?
> >
> > Finally, thanks for the additional experiments --- I think they really strengthen the evaluation and I appreciate the effort.

---

> ### Author Response · Authors · 2025-11-19
>
> Thank you for the appreciation of our additional experiments that strengthen the evaluation, and for acknowledging that our responses addressed your original concerns.
>
>
> > Thanks for the response. I feel like this addresses Weaknesses 3/4 reasonably well. If you can commit to making this more clear in the paper, I'd be satisfied here.
>
> Thank you for the follow-up. We now clarified both points directly in the paper. To make our response clear to address weakness 3, we have now added one sentence in Lines 252-253, “These metrics are calculated against the ground-truth risk labels, i.e., the harmful indices which are described in the individual subsection of each task in Section 2.”  This makes the evaluation procedure unambiguous.
>
> To address Weakness 4, we added a clarifying sentence at Line 121: “Additionally, in Section A.2, we analyze whether our dataset exhibits information leakage due to being generated by the same pipeline, and we find no evidence that such leakage affects our results.” We believe these revisions make both points substantially clearer in the main paper.
>
>
>
>
> > I should be clear that I think stateful decomposition attacks are interesting. However, I think the paper is not nearly clear enough about this scope up front.
>
> > Thanks for the clarity here. I think this is convincing. I'd recommend making your claims more clearly grounded in this evidence to avoid misinterpretation. I would also be more comfortable if the claim is more carefully scoped to apply to stateful decomposition attacks within a single context.
>
> Thank you for this helpful suggestion. We agree that clearly scoping our claims to the stateful setting is important to avoid misinterpretation. While we already mention “stateful” in the first sentence of the Abstract (Lines 11–12), we have revised the paper to make this scope even more explicit. Specifically, we now add “in the same context” to the first sentence of the Abstract and reiterate that our findings apply to stateful decomposition attacks in Lines 28 and mention it again that our dataset is for stateful decomposition attack in Line 96. These additions ensure that readers are immediately aware of the intended scope and that our claims are appropriately grounded in the presented evidence. We hope this addresses the concern.
>
>
> > The discussion could more clearly lay out future work on extending to stateless attacks across distinct contexts.
>
> Thank you for this suggestion. We now move the future work section on stateless attacks, originally in Appendix E.1, to the Discussion section. Specifically, line 505, highlighted in yellow.
>
>
> >  For example, afaict, the attack you reference from Anthropic was an example of a stateless attack where the decomposition between attacks occurred across contexts. Is there something that I've missed?
>
> Whether it is stateful or not is not clearly reported in the Anthropic report, but this attack can be inherently seen as stateful because the orchestration layer maintained and updated attack state across phases while sequentially conditioning the next subtask on the prior outputs. Even though our work focuses on monitoring model inputs, the same techniques can be applied to the outputs of an orchestration/manager model that interacts with individual agents, since those sequential outputs can serve as the monitoring targets for our lightweight sequential monitor. However, we leave this future work to explore. Additionally, suppose Anthropic applies the proposed solution we discuss in the future work section (specifically, line 505, highlighted in yellow). In that case, our work can also be directly applied there, as similar tasks can be grouped in the same context, allowing our monitor to safeguard against them.
>
> > I understand the desire to compare across appropriate settings, however I'd encourage you to compare with prompt-optimization for more capable models to understand the limits of performance. This would help understand if, e.g., distilling a prompt-optimized larger model through finetuning could be valuable. I'm sensitive to concerns about cost, but I want to suggest the direction as a valuable one.
>
> Thank you for the suggestion. We think this is indeed an interesting direction, but it wouldn’t change the main claim we make. We are excited for future work to explore this.

---

> > ### Author Response · Authors · 2025-11-19
> >
> > > RE: judge validation, please make sure that this is clearly communicated in the paper. As a followup question, can you address the possibility of distribution shift in your queries reducing the alignment between the LLM judge and human evaluations?
> >
> > Thank you for this follow-up question. In response, we now include the full judge prompt in Figure 19. We have also restructured Appendix A.2 to clearly delineate (1) Duplicate Detection, (2) Manual Validation of GPT-5 Judgments, (3) Evaluation, and (4) Results.
> >
> > Regarding the concern about potential distribution shift reducing alignment between the LLM judge and human evaluators: in our setting, this risk is mitigated because GPT-5 is only applied to a narrow top 5 candidate set retrieved via embedding similarity, meaning it operates on decomposition pairs that are already close in semantic space. Moreover, our manual validation, reviewing all 62 flagged pairs and 45 unflagged samples, shows strong agreement between GPT-5 and human annotations, with only a small number of false positives (<5) and no observed false negatives. These findings suggest that the distribution shift is unlikely to materially affect our redundancy analysis. We now discuss this explicitly in the “Manual Validation of GPT-5 Judgment” paragraph.
> >
> >
> > Thank you again for the thoughtful feedback and for engaging so deeply with the work. Please let us know if you have any other questions that we can address. If not, we sincerely hope you can consider increasing your rating to 6 (Borderline Accept), as we believe that our contribution is significant to the safety community that aims to guard against sophisticated misuse.

---

> > > ### Comment · Reviewer_svdc · 2025-11-19
> > > **Thank you for engaging**
> > >
> > > Thank you, I appreciate your efforts and plans to improve the paper. I'm now much more confident in the quality of the paper and have revised my review to an accept (8) with confidence 4 to reflect that.

---

> > > > ### Author Response · Authors · 2025-11-19
> > > >
> > > > Thank you for raising the score to 8. We appreciate your recognition of our contribution.

---

### Official Review · Reviewer_mdsg · 2025-11-04

**Soundness:** 4
**Presentation:** 4
**Contribution:** 3
**Rating:** 6
**Confidence:** 3

**Summary:**

The submission produces a dataset ("DecomposedHarm") of highly-effective LLM jailbreaks that rely on decomposing a harmful query into multiple benign-seeming subtasks. Careful experiments show that LLM-based sequential monitors (classifiers of the subtask/prompt sequence) can accurately identify when these subtasks will begin to induce a harmful output. To address the financial cost and latency of such LLM-based monitors, prompt engineering is used to enhance the performance of lightweight LLM sequential monitors (beyond the performances of more sophisticated and expensive models). Notably, even when additional adversarial strategies are added to the decomposition attack, the lightweight sequential monitors robustly discriminate between benign and harmful subtask sequences.

**Strengths:**

**Originality** The work introduces a new dataset (DecomposedHarm), and a simple, practical method for addressing decomposition attacks.

**Quality** DecomposedHarm addresses a variety of potential jailbreak settings, from image generation to agentic LLM tasks. Comparisons include strong baselines like Llama Guard, which is greatly outperformed. Adversarial evaluation provides additional evidence of the proposed method's benefits. Decompositions are verified by human reviewers. The limitations are thorough and helpful.

**Clarity** The paper is very well written, with clear figures and tables.

**Significance** The submission addresses an urgent problem of practical importance. The dataset DecomposedHarm will facilitate future research in this area.

**Weaknesses:**

The submission has no significant weaknesses.

In the limitations section, perhaps emphasize that decomposition is just one of many attack strategies, and decomposition could potentially play a role in building composite attacks (e.g. with genetic algorithms) stronger than those explored in the submission.

**Questions:**

1. Line 178: why are the harmful indices the last indices? Couldn't the image become harmful before the last subtask?
2. Did you employ any checks for redundancy in the LLM generated examples that populate DecomposedHarm? A little redundancy is okay, but not if you have overlap between your validation and test sets.
3. Adding a newer model (Gemini 2.5, Sonnet 4.5, and/or GPT 5) as a reference in Table 1 could boost the relevance of the analysis and clarify what the frontier is. Relatedly, a newer compact model (Haiku?) could be a strong candidate for prompt-engineering.
4. In Table 1, GPT 4o without any optimization appears cost effective and performant. It would be interesting to see how the references perform with a prompt (latency and F1).
5. Table 1: The GPT 4o mini F1 does not seem to match Table 6’s.
6. Line 351: there seems to be a typo here – is “o3-mini” supposed to be “GPT 4o”?

---

> ### Author Response · Authors · 2025-11-16
>
> Thank you for reviewing our work and for the positive rating. We answer your questions below:
>
> > Weakness: In the limitations section, perhaps emphasize that decomposition is just one of many attack strategies, and decomposition could potentially play a role in building composite attacks (e.g. with genetic algorithms) stronger than those explored in the submission.
>
> Thanks for this suggestion. We added this in Lines 498 and 499, highlighted in yellow.
>
> > Q1: Line 178: why are the harmful indices the last indices? Couldn't the image become harmful before the last subtask?
>
> In our image-task generation pipeline, the harmful content is always introduced in the final subprompt, specifically, by specifying a race or nationality. Without this last instruction, the scene contains no identity-based or stereotypical elements. Therefore, the harmful indices correspond to the very last subprompt for our text-to-image tasks.
>
> > Q2: Did you employ any checks for redundancy in the LLM generated examples that populate DecomposedHarm? A little redundancy is okay, but not if you have overlap between your validation and test sets.
>
> Thank you for raising this point. To answer your question, we conducted additional analyses to check for redundancy or leakage between the generated examples in DecomposedHarm. As detailed in Appendix A.2 (page 16), we removed all duplicates and re-ran the evaluation. Monitoring performance changed only trivially—well within one standard error of the original estimates. This is expected given that the test set contains very few duplicates. These results indicate that redundancy does not materially affect our conclusions.
>
> > Q3: Adding a newer model (Gemini 2.5, Sonnet 4.5, and/or GPT 5) as a reference in Table 1 could boost the relevance of the analysis and clarify what the frontier is. Relatedly, a newer compact model (Haiku?) could be a strong candidate for prompt-engineering.
>
> Thank you! Following your suggestion, we added GPT-5, as shown in Table 1 on Page 7. Our lightweight, optimized monitor (GPT-4o-mini) still outperformed it.
>
> > Q4: In Table 1, GPT 4o without any optimization appears cost effective and performant. It would be interesting to see how the references perform with a prompt (latency and F1).
>
> Thank you for this comment. As shown in Figure 5(a) (now Figure 4(a) after rebuttal revision), GPT-4o is already too costly to be considered a practical monitoring option, which is why we did not include it in the prompt-optimization study. Our goal is to compare baselines and lightweight monitors under realistic cost constraints. For reference models such as GPT-4o, o3-mini, and GPT-5, prompt optimization would require substantially longer and more complex prompts, further increasing inference cost and latency beyond their already high default settings. Since these models are not feasible candidates for deployment as monitors, optimizing them would not represent a realistic comparison.
>
> > Q5: Table 1: The GPT 4o mini F1 does not seem to match Table 6’s.
>
> Thank you for the comment. In Table 1, we report the performance of GPT-4o-mini using optimized prompts. In contrast, Table 6 (now Table 8 because we added 2 other tables during the rebuttal) presents results using the default binary prompt within the sequential framework, as its purpose is to compare single-input monitoring with the sequential framework.
>
> > Q6: Line 351: there seems to be a typo here – is “o3-mini” supposed to be “GPT 4o”?
>
> Thank you. We fixed it and highlighted it in yellow.
>
> Additional note: Given that Anthropic’s recent report on an AI-orchestrated cyber-espionage campaign [https://www.anthropic.com/news/disrupting-AI-espionage] explicitly uses decomposition attacks, “broke down their attacks into small, seemingly innocent tasks that Claude would execute without being provided the full context of their malicious purpose”, we think our work is highly relevant, realistic, and well-positioned to address this increasingly effective misuse strategy.
>
> Please let us know if you have any other questions that we can address. If not, we sincerely hope you can consider increasing your rating to 8 (accept), as we believe that our contribution is significant to the safety community that aims to guard against sophisticated, realistic misuse.

---

> > ### Comment · Reviewer_mdsg · 2025-11-23
> >
> > Thank you for the responses. I agree that this work is timely!
> >
> > I am interested in discussing one point further. Specifically, it appears in Table 1 that GPT 4o has a latency similar to monitor models Llama-3.1-8B and GPT 4o mini. Is this a typo? If not, could you please explain why you say
> >
> > > GPT-4o is already too costly to be considered a practical monitoring option,
> >
> > but then consider Llama-3.1-8B to not be too costly despite its having similar latency and financial cost?

---

> ### Author Response · Authors · 2025-11-23
>
> Thanks for the acknowledgement that our work is timely.
>
> We note that GPT-4o is approximately 1.5 times more expensive than Llama-3.1-8B (4.17 vs. 2.80 in Table 1), and its input tokens are around 13.8 times more costly (2.5 vs. 0.18 in Figure 4(a)). Optimizing GPT-4o would require substantially increasing the number of input tokens, which would even further widen the cost gap between GPT-4o and Llama-3.1-8B in practice (ie more than what we showed in Table 1). Taken together, these considerations are why we state that “GPT-4o is already too costly to be considered a practical monitoring option.” While the two models have indeed similarly low latency, the substantial cost difference is the primary reason that makes GPT-4o a less ideal choice for deployment-time monitoring.
>
> Thank you for this follow-up! Please let us know if you have any other questions that we can address. If not, we sincerely hope you can consider increasing your rating to 8 (accept), as we believe that our contribution is timely and significant to the safety community that aims to guard against sophisticated, realistic misuse.

---

> > ### Author Response · Authors · 2025-11-28
> >
> > Thank you for raising your score to 8. We appreciate your recognition of our contribution.

---

### Author Response · Authors · 2025-11-23

We are pleased that all reviewers have provided positive assessments of our work by raising scores to 8, 8, 8, and 6 during the rebuttal. We want to note that we highlighted our revisions in yellow in our rebuttal version of the paper to help reviewers and ACs easily identify the changes. These highlights will be removed in the camera-ready version.

Additionally, we are grateful that the reviewers recognized the strengths of our work, including: addressing a realistic and important problem, decomposition attacks; creating a valuable dataset, DecomposedHarm, for evaluating defenses against such attacks; proposing a surprisingly simple yet robust method that can efficiently guard against decomposition attacks; and conducting extensive adversarial pressure tests demonstrating the robustness of our defense method.

Lastly, we want to note that given that Anthropic’s recent report on an AI-orchestrated cyber-espionage campaign [https://www.anthropic.com/news/disrupting-AI-espionage] explicitly uses decomposition attacks, “broke down their attacks into small, seemingly innocent tasks that Claude would execute without being provided the full context of their malicious purpose”, we think that our work is highly relevant, realistic, and well-positioned to address this increasingly effective misuse strategy.

We deeply appreciate the rigorous and insightful review process.

---

### Meta-Review · Area_Chair_Ypcq · 2026-01-02

**Summary:**

The reviewers have raised the following major concerns:
1. The paper overclaims the work's representativeness and contributions. (svdc)
2. The experimental setup needs further justification. (svdc)
3. The computational cost is high. (JNS6)
4. The defense robustness may stem from prompting rather than method design. (JNS6, eW3J)
5. No adaptive attacks are considered. (JNS6)
6. Need more details about the datasets. (eW3J)

**Reviewer Concerns:**

Concerns addressed by rebuttal:
(1) (2) (4) (5) (6)

Outstanding concerns:
(3)

**Reviewer Scores:**

- mdsg: No significant concerns are raised.
- svdc: After extensive exchange, most of the reviewer's concerns are addressed. They would likely increase their score.
- JNS6: The rebuttal addresses most of the reviewer's comments; They would likely increase their score.
- eW3J: The rebuttal addresses most of the reviewer's comments; They would likely increase their score.

---

### Decision · Program_Chairs · 2026-01-26

Accept (Poster)